# MixTEA: Semi-supervised Entity Alignment with Mixture Teaching

**Feng Xie, Xin Song, Xiang Zeng, Xuechen Zhao, Lei Tian, Bin Zhou**[(✉)]**, Yusong Tan**
College of Computer, National University of Defense Technology
{xiefeng,songxin,zengxiang,zhaoxuechen,leitian129,ystan}@nudt.edu.cn
✉ binzhou@nudt.edu.cn

## Abstract

Semi-supervised entity alignment (EA) is a practical and challenging task because of the lack of adequate labeled mappings as training data. Most works address this problem by generating pseudo mappings for unlabeled entities. However, they either suffer from the erroneous (noisy) pseudo mappings or largely ignore the uncertainty of pseudo mappings. In this paper, we propose a novel semi-supervised EA method, termed as MixTEA, which guides the model learning with an end-to-end mixture teaching of manually labeled mappings and probabilistic pseudo mappings. We firstly train a student model using few labeled mappings as standard. More importantly, in pseudo mapping learning, we propose a bi-directional voting (BDV) strategy that fuses the alignment decisions in different directions to estimate the uncertainty via the joint matching confidence score. Meanwhile, we also design a matching diversity-based rectification (MDR) module to adjust the pseudo mapping learning, thus reducing the negative influence of noisy mappings. Extensive results on benchmark datasets as well as further analyses demonstrate the superiority and the effectiveness of our proposed method.

## 1 Introduction

Entity alignment (EA) is a task at the heart of integrating heterogeneous knowledge graphs (KGs) and facilitating knowledge-driven applications, such as question answering, recommender systems, and semantic search (Gao et al., 2018). Embedding-based EA methods (Chen et al., 2017; Wang et al., 2018; Sun et al., 2020a; Yu et al., 2021; Xin et al., 2022a) dominate current EA research and achieve promising alignment performance. Their general pipeline is to first encode the entities from different KGs as embeddings (latent representations) in a uni-space, and then find the most likely counterpart for each entity by performing all pairwise comparison. However, the pre-aligned mappings (i.e.,

training data) are oftentimes insufficient, which is challenging for supervised embedding-based EA methods to learn informative entity embeddings. This happens because it is time-consuming and labour-intensive for technicians to manually annotate entity mappings in the large-scale KGs.

To remedy the lack of enough training data, some existing efforts explore alignment signals from the cheap and valuable unlabeled data in a semi-supervised manner. The most common semi-supervised EA solution is using the self-training strategy, i.e., iteratively generating pseudo mappings and combining them with labeled mappings to augment the training data. For example, Zhu et al. (2017) propose IPTransE which involves an iterative process of predicting on unlabeled data and then treats the predictions above an elaborate threshold (confident predictions) as pseudo mappings for retraining. To further improve the accuracy of pseudo mappings, Sun et al. (2018) design a heuristic editing method to remove wrong alignment by considering one-to-one alignment constraint, while Mao et al. (2020) and Cai et al. (2022) utilize a bi-directional iterative strategy to determine pseudo mapping if and only if the two entities are mutually nearest neighbors of each other. Despite the encouraging results, existing semi-supervised EA methods still face the following problems: (1) **Uncertainty of pseudo mappings**. Prior works have largely overlooked the uncertainty of pseudo mappings during semi-supervised training. Revisiting the self-training process, the generation of pseudo mappings is either black or white, i.e., an entity pair is either determined as a pseudo mapping or not. While in fact, different pseudo mappings have different uncertainties and contribute differently to model learning (Zheng and Yang, 2021). (2) **Noisy pseudo mapping learning**. The performance of semi-supervised EA methods depends heavily on the quality of pseudo mappings, while these pseudo mappings inevitably

contain much noise (i.e., False Positive mappings). Even worse, adding them into the training data would misguide the subsequent training process, thus causing error accumulation and further hurting the alignment performance.

To tackle the aforementioned limitations, in this paper, we propose a simple yet effective semi-supervised EA solution, termed as MixTEA. To be specific, our method is based on a Teacher-Student architecture (Tarvainen and Valpola, 2017), which aims to generate pseudo mappings from a gradually evolving teacher model and guides the learning of a student model with a mixture teaching of labeled mappings and pseudo mappings. We explore the uncertainty of pseudo mappings via probabilistic pseudo mapping learning rather than directly adding "reliable" pseudo mappings into the training data, which lends us to flexibly learn from pseudo mappings with different uncertainties. To achieve that, we propose a bi-directional voting (BDV) strategy that utilizes the consistency and confidence of alignment decisions in different directions to estimate the uncertainty via the *joint matching confidence score* (converted to matching probability after a softmax). Meanwhile, a matching diversity-based rectification (MDR) module is designed to adjust the pseudo mapping learning, thus reducing the influence of noisy mappings. Our contributions are summarized as follows:

(**I**) We propose a novel semi-supervised EA framework, termed as MixTEA[1], which guides the model's alignment learning with an end-to-end mixture teaching of manually labeled mappings and probabilistic pseudo mappings.

(**II**) We introduce a bi-directional voting (BDV) strategy which utilizes the alignment decisions in different directions to estimate the uncertainty of pseudo mappings and design a matching diversity-based rectification (MDR) module to adjust the pseudo mapping learning, thus reducing the negative impacts of noise mappings.

(**III**) We conduct extensive experiments and thorough analyses on benchmark datasets OpenEA (Sun et al., 2020b). The results demonstrate the superiority and effectiveness of our proposed method.

## 2 Related Works

### 2.1 Embedding-based Entity Alignment

While the recent years have witnessed the rapid development of deep learning techniques, embedding-

based EA approaches obtain promising results. Among them, some early studies (Chen et al., 2017; Sun et al., 2017) are based on the knowledge embedding methods, in which entities are embedded by exploring the fine-grained relational semantics. For example, MTransE (Chen et al., 2017) applies TransE (Bordes et al., 2013) as the KG encoder to embed different KGs into independent vector spaces and then conducts transitions via designed alignment modules. However, they need to carefully balance the weight between the encoder and alignment module in one unified optimization problem. Due to the powerful structure learning capability, Graph Neural Networks (GNNs) like GCN (Kipf and Welling, 2017) and GAT (Veličković et al., 2018) have been employed as the encoder with Siamese architecture (i.e., shared-parameter) for many embedding-based models. GCN-Align (Wang et al., 2018) applies Graph Convolution Network (GCN) for the first time to capture neighborhood information and embed entities into a unified vector space, but it suffers from the structural heterogeneity of different KGs. To mitigate this issue and improve the structure learning, AliNet (Sun et al., 2020a) adopts multi-hop aggregation with a gating mechanism to expand neighborhood ranges for better structure modeling, and KE-GCN (Yu et al., 2021) combines GCN and knowledge embedding methods to jointly capture the rich structural features and relation semantics of entities. More recently, IMEA (Xin et al., 2022a) designs a Transformer-like architecture to encode multiple structural contexts in a KG while capturing alignment interactions across different KGs.

In addition, some works further improve the EA performance by introducing the side information about entities, such as entity names (Zhang et al., 2019), attributes (Liu et al., 2020), and literal descriptions (Yang et al., 2019). Afterward, a series of methods were proposed to integrate knowledge from different modalities (e.g., relational, visual, and numerical) to obtain joint entity representation for EA (Chen et al., 2020; Liu et al., 2021; Lin et al., 2022). However, these discriminative features are usually hard to collect, noise polluted, and privacy sensitive (Pei et al., 2022).

### 2.2 Semi-supervised Entity Alignment

Since the manually labeled mappings used for training are usually insufficient, many semi-supervised EA methods have been proposed to take advan-

---

[1] https://github.com/Xiefeng69/MixTEA

tage of labeled mappings and the large amount of unlabeled data for alignment, which can provide a more practical solution in real scenarios. The mainstream solutions focus on iteratively generating pseudo mappings to compensate for the lack of training data. IPTransE (Zhu et al., 2017) applies threshold filtering-based self-training to yield pseudo mappings but it fails to obtain satisfactory performance since it brings much noise data, which would misguide the subsequent training. Besides, it is also hard to determine an appropriate threshold to select "confident" pseudo mappings. KDCoE (Chen et al., 2018) performs co-training of KG embedding model and literal description embedding model to gradually propose new pseudo mappings and thus enhance the supervision of alignment learning for each other. To further improve the quality of pseudo mappings, BootEA (Sun et al., 2018) designs an editable strategy based on the one-to-one matching rule to deal with matching conflicts and MRAEA (Mao et al., 2020) proposes a bi-directional iterative strategy which imposes a mutually nearest neighbor constraint. Inspired by the success of self-training, RANM (Cai et al., 2022) proposes a relation-based adaptive neighborhood matching method for entity alignment and combines a bi-directional iterative co-training strategy, making become a natural semi-supervised model. Moreover, CycTEA (Xin et al., 2022b) devises an effective ensemble framework to enable multiple alignment models (called aligners) to exchange their reliable entity mappings for more robust semi-supervised training, but it requires high complementarity among different aligners.

Additionally, other effective semi-supervised EA methods, such as SEA (Pei et al., 2019), RAC (Zeng et al., 2021), GAEA (Xie et al., 2023), focus on introducing specific loss terms (e.g., reconstruction loss, contrastive loss) via auxiliary tasks.

## 3 Problem Statement

A knowledge graph (KG) is formalized as $\mathcal{G} = (\mathcal{E}, \mathcal{R}, \mathcal{T})$, where $\mathcal{E}$ and $\mathcal{R}$ refer to the set of entities and the set of relations, respectively. $\mathcal{T} = \mathcal{E} \times \mathcal{R} \times \mathcal{E} = \{(h, r, t)|h, t \in \mathcal{E} \wedge r \in \mathcal{R}\}$ is the set of triples, where $h$, $r$, and $t$ denote head entity (subject), relation, tail entity (object), respectively. Given a source KG $\mathcal{G}_s = (\mathcal{E}_s, \mathcal{R}_s, \mathcal{T}_s)$, a target KG $\mathcal{G}_t = (\mathcal{E}_t, \mathcal{R}_t, \mathcal{T}_t)$, and a small set of pre-aligned mappings (called training data) $\mathcal{S} = \{(e_s, e_t)|e_s \in \mathcal{E}_s \wedge e_t \in \mathcal{E}_t \wedge e_s \equiv e_t\}$, where $\equiv$ means equivalence

relationship, entity alignment (EA) task pairs each source entity $e_i \in \mathcal{E}_s$ via nearest neighbor (NN) search to identify its corresponding target entity $e_j \in \mathcal{E}_t$:

$$e_j = \arg \min_{\tilde{e}_j \in \mathcal{E}_t} d(e_i, \tilde{e}_j) \tag{1}$$

where $d(\cdot)$ denotes distance metrics (e.g., Manhattan or Euclidean distance). Moreover, to mitigate the inadequacy of training data, semi-supervised EA methods make effort to explore more potential alignment signals over the vast unlabeled entities, i.e., $\hat{\mathcal{E}}_s$ and $\hat{\mathcal{E}}_t$, which denote the unlabeled entity set of source KG and target KG, respectively.

## 4 Proposed Method

In this section, we present our proposed semi-supervised EA method, called MixTEA, in Figure 1. MixTEA follows the teacher-student training scheme. The teacher model is performed to generate probabilistic pseudo mappings on unlabeled entities and student model is trained with an end-to-end mixture teaching of manually labeled mappings and probabilistic pseudo mappings. Compared to previous methods that require filtering pseudo mappings via thresholds or constraints, the end-to-end training gradually improves the quality of pseudo mappings, and the more and more accurate pseudo mappings in turn benefit EA training.

### 4.1 KG Encoder

We first introduce the KG encoder (denoted as $f(; \theta)$) which utilizes neighborhood structures and relation semantics to embed entities from different KGs into a unified vector space. We randomly initialize the trainable entity embeddings $\mathbf{H}^{ent} \in \mathbb{R}^{(|\mathcal{E}_s|+|\mathcal{E}_t|) \times d_e}$ and relation embeddings $\mathbf{H}^{rel} \in \mathbb{R}^{|\mathcal{R}_s \cup \mathcal{R}_t| \times d_r}$, where $d_e$ and $d_r$ are the dimension of entities and relations, respectively.

**Structure modeling.** Structural features are crucial since equivalent entities tend to have similar neighborhood contexts. Besides, leveraging multi-range neighborhood structures is capable of providing more alignment evidence and mitigating the structural heterogeneity issue. In this work, we apply Graph Attention Network (GAT) (Veličković et al., 2018) to allow an entity to selectively aggregate its surrounding information via attentive mechanism and we then recursively capture multi-range structural features by stacking $L$ layers:

$$\mathbf{h}_{e_i}^{(l)} = \sigma \left( \sum_{e_j \in \mathcal{N}_{e_i}} \alpha_{ij} \mathbf{W}_g \mathbf{h}_{e_j}^{(l-1)} \right) \tag{2}$$

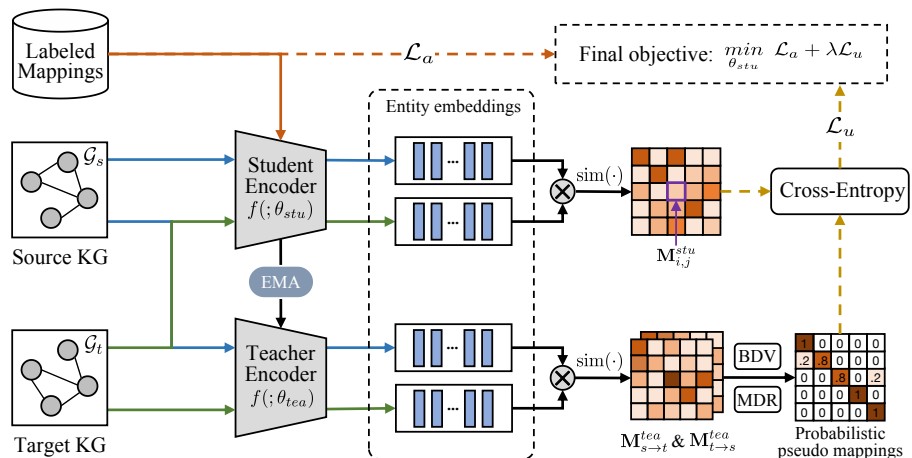

Figure 1: The overall of our proposed MixTEA, which consists of two KG encoders, called student model and teacher model. We obtain entity embeddings via the KG encoder. Both labeled mappings and probabilistic pseudo mappings are used to train the student model. The final student model is used for alignment inference.

$$\alpha_{ij} = \frac{\exp(\sigma(\mathbf{a}^\top[\mathbf{W}_g\mathbf{h}_{e_i} \oplus \mathbf{W}_g\mathbf{h}_{e_j}]))}{\sum_{e_z \in \mathcal{N}_{e_i}} \exp(\sigma(\mathbf{a}^\top[\mathbf{W}_g\mathbf{h}_{e_i} \oplus \mathbf{W}_g\mathbf{h}_{e_z}]))} \quad (3)$$

where $\top$ represents transposition, $\oplus$ means concatenation, $\mathbf{W}_g$ and $\mathbf{a}$ are the layer-specific transformation parameter and attention transformation vector, respectively. $\mathcal{N}_{e_i}$ means the neighbor set of $e_i$ (including $e_i$ itself by adding a self-connection), and $\alpha_{ij}$ indicates the learned importance of entity $e_j$ to entity $e_i$. $\mathbf{H}^{(l)}$ denotes the entity embedding matrix at $l$-th layer with $\mathbf{H}^{(0)} = \mathbf{H}^{ent}$. $\sigma(\cdot)$ is the nonlinear function and we use ELU here.

**Relation modeling.** Relation-level information which carries rich semantics is vital to align entities in KGs because two equivalent entities may share overlapping relations. Considering that relation directions, i.e., outward ($e_i \rightarrow e_j$) and inward ($e_i \leftarrow e_j$), have delicate impacts on characterizing the given target entity $e_i$, we use two mean aggregators to gather outward and inward relation semantics separately to provide supplementary features for heterogeneous KGs:

$$\mathbf{h}_{e_i}^{r+} = \frac{1}{|\mathcal{N}_{e_i}^{r+}|} \sum_{r \in \mathcal{N}_{e_i}^{r+}} \mathbf{h}_r^{rel} \quad (4)$$

$$\mathbf{h}_{e_i}^{r-} = \frac{1}{|\mathcal{N}_{e_i}^{r-}|} \sum_{r \in \mathcal{N}_{e_i}^{r-}} \mathbf{h}_r^{rel} \quad (5)$$

where $\mathcal{N}_{e_i}^{r+}$ and $\mathcal{N}_{e_i}^{r-}$ are the sets of outward and inward relations of entity $e_i$, respectively.

**Weighted concatenation.** After capturing the contextual information of entities in terms of neighborhood structures and relation semantics, we con-

catenate intermediate features for entity $e_i$ to obtain the final entity representation:

$$\mathbf{h}_{e_i} = \mathop{\oplus}_{k \in K} \left[ \frac{\exp(\mathbf{w}_k)}{\sum \exp(\mathbf{w})} \cdot \mathbf{h}_{e_i}^k \right] \quad (6)$$

where $K = \{(1), ..., (L), r+, r-\}$ and $\mathbf{w} \in \mathbb{R}^{|K|}$ is the trainable attention vector to adaptively control the flow of each feature. We feed $\mathbf{w}$ to a softmax before multiplication to ensure that the normalized weights sum to 1.

## 4.2 Alignment Learning with Mixture Teaching

In the following, we will introduce mixture teaching, which is reached by the supervised alignment learning and probabilistic pseudo mapping learning in an end-to-end training manner.

**Teacher-student architecture.** Following Mean Teacher (Tarvainen and Valpola, 2017), we build our method which consists of two KG encoders with identical structure, called student model $f(;\theta_{stu})$ and teacher model $f(;\theta_{tea})$, respectively. The student model constantly updates its parameters supervised by the manually labeled mappings as standard and the teacher model is updated via the exponential moving average (EMA) (Tarvainen and Valpola, 2017) weights of the student model. Moreover, the student model also learns from the pseudo mappings generated by the teacher model to further improve its performance, in which the uncertainty of pseudo mappings is formalized as calculated matching probabilities. Specifically, we update the teacher model as follows:

$$\theta_{tea} \leftarrow m\theta_{tea} + (1-m)\theta_{stu}, m \in [0, 1) \quad (7)$$

where $\theta$ denotes model weights, and $m$ is a preset momentum hyperparameter that controls the teacher model to update and evolve smoothly.

**Supervised alignment learning.** In order to make equivalent entities close to each other and unmatched entities pull away from each other in a unified space, we apply a margin-based alignment loss (Wang et al., 2018; Mao et al., 2020; Yu et al., 2021) supervised by pre-aligned mappings:

$$
\begin{aligned}
\mathcal{L}_a = \sum_{(e_s,e_t)\in\mathcal{S}} \sum_{(\bar{e}_s,\bar{e}_t)\in\bar{\mathcal{S}}} & [||\mathbf{h}_{e_s} - \mathbf{h}_{e_t}||_2 \\
& + \rho - ||\mathbf{h}_{\bar{e}_s} - \mathbf{h}_{\bar{e}_t}||_2]_+
\end{aligned} \quad (8)
$$

where $\rho$ is a hyperparameter of margin, $[x]_+ = \max\{0, x\}$ is to ensure non-negative output, $\bar{\mathcal{S}}$ denotes the set of negative entity mappings, and $||\cdot||_2$ means L2 distance (Euclidean distance). Negative mappings are sampled according to the cosine similarity of two entities (Sun et al., 2018).

**Probabilistic pseudo mapping learning.** As mentioned above, the teacher model is responsible for generating probabilistic pseudo mappings for the student model to provide more alignment signals and thus enhance the alignment performance. Benefiting from the EMA update, the predictions of the teacher model can be seen as an ensemble version of the successive student models' predictions. Therefore it is more robust and stable for pseudo mapping generation. Moreover, bi-directional iterative strategy (Mao et al., 2020) reveals the asymmetric nature of alignment directions (i.e., source-to-target and target-to-source), which can produce pseudo mappings based on the mutually nearest neighbor constraint. Inspired by this, we propose a bi-directional voting (BDV) strategy which fuses alignment decisions in each direction to yield more comprehensive pseudo mappings and model their uncertainty via the *joint matching confidence score*. Concretely, after encoding, we can first obtain the similarity matrix by performing pairwise similarity calculation between the unlabeled source and target entities as follows:

$$
\mathbf{M}_{s\to t}^{tea} = \text{sim}(\hat{\mathcal{E}}_s, \hat{\mathcal{E}}_t, \theta_{tea}) \in \mathbb{R}^{|\hat{\mathcal{E}}_s|\times|\hat{\mathcal{E}}_t|} \quad (9)
$$

where $\text{sim}(\cdot)$ denotes cosine similarity function. $\mathbf{M}_{s\to t}^{tea}$ and $\mathbf{M}_{t\to s}^{tea}$ represent similarity matrices in different directions between source and target entities, and $\mathbf{M}_{t\to s}^{tea}$ is the transposition of $\mathbf{M}_{s\to t}^{tea}$ (i.e., $\mathbf{M}_{t\to s}^{tea} = (\mathbf{M}_{s\to t}^{tea})^\top$). Next, for each matrix, we pick up the entity pair which has the maximum

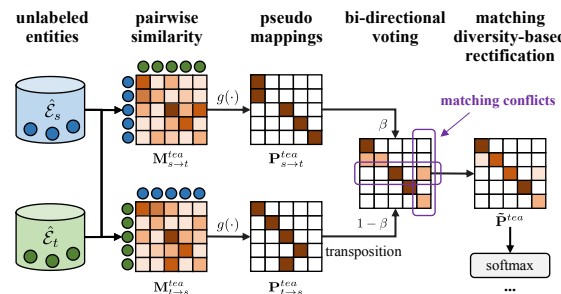

Figure 2: The illustration of the process of generating the probabilistic pseudo mapping matrix from two alignment directions. We assume that entity pairs on the diagonal are correct mappings and that the default alignment direction for inference is from source to target.

predicted similarity in each row as the pseudo mapping and then we combine the results of the pseudo mappings in different directions weighted by their last *Hit@1* scores on validation data to obtain the final pseudo mapping matrix:

$$
\mathbf{P}^{tea} = \beta \cdot g(\mathbf{M}_{s\to t}^{tea}) + (1 - \beta) \cdot g(\mathbf{M}_{t\to s}^{tea})^\top \quad (10)
$$

$$
\beta = \frac{\text{valid}(\mathbf{M}_{s\to t}^{tea})}{\text{valid}(\mathbf{M}_{s\to t}^{tea}) + \text{valid}(\mathbf{M}_{t\to s}^{tea})} \quad (11)
$$

$$
g(\mathbf{M}) = [m_{i,j}] = \begin{cases} 1, & \text{if } j = \text{arg}max_j\,\mathbf{M}_{i,j} \\ 0, & \text{otherwise} \end{cases} \quad (12)
$$

where $g(\cdot)$ is the function that converts the similarity matrix to a one-hot matrix (i.e., the only position with a value 1 at each row of the matrix indicates the pseudo mapping). In this manner, we arrive at the final pseudo mapping matrix $\mathbf{P}^{tea}$ generated by the teacher model, in which each pseudo-mapping is associated with a joint matching confidence score (the higher the joint matching confidence, the less the uncertainty). Different from the bi-directional iterative strategy, we use the voting consistency and matching confidence of alignment decisions in different directions to facilitate uncertainty estimation. Specifically, given an entity pair $(\hat{e}_i, \hat{e}_j)$, its confidence $\mathbf{P}_{i,j}^{tea}$ is 1 when and only when both directions unanimously vote this entity pair as a pseudo mapping, otherwise its confidence is in the interval (0,1) when only one direction votes for it and 0 when no direction votes for it (i.e., this entity pair will not be regarded as a pseudo mapping).

In addition, the ideal predictions of EA need to satisfy the one-to-one matching constraint (Suchanek et al., 2011; Sun et al., 2018), i.e., a source entity can be matched with at most one target entity, and vice versa. However, the joint

decision voting process inevitably yields matching conflicts due to the existence of erroneous (noisy) mappings. Inspired by Gal et al. (2016), we further propose a matching diversity-based rectification (MDR) module to adjust the pseudo mapping learning, thus mitigating the influence of noisy mappings dynamically. We denote $\mathbf{M}^{stu}$ (i.e., $\mathbf{M}_{i,j}^{stu} = \text{sim}(\hat{e}_i, \hat{e}_j; \theta_{stu})$) as the similarity matrix calculated based on the student model and define a Cross-Entropy (CE) loss between $\mathbf{M}^{stu}$ and $\mathbf{P}^{tea}$ rectified by matching diversity:

$$\tilde{\mathbf{P}}_{i,j}^{tea} = \frac{\mathbf{P}_{i,j}^{tea}}{\sum \mathbf{P}_{i:}^{tea} + \sum \mathbf{P}_{:j}^{tea} - \mathbf{P}_{i,j}^{tea}} \quad (13)$$

$$\mathcal{L}_u = \sum_{i=1}^{|\hat{\mathcal{E}}_s|} \text{Cross-Entropy}(\mathbf{M}_{i:}^{stu}, \tilde{\mathbf{P}}_{i:}^{tea}) \quad (14)$$

where $\tilde{\mathbf{P}}^{tea}$ denotes the rectified pseudo mapping matrix. To be specific, the designed rectification term (Eq. (13)) measures how much a potential pseudo mapping deviates (in terms of joint matching confidence score) from other competing pseudo mappings in $\mathbf{P}_{i:}^{tea}$ and $\mathbf{P}_{:j}^{tea}$. The larger the deviation, the greater the penalty for this pseudo mapping, and vice versa. Notably, both $\mathbf{M}^{stu}$ and $\tilde{\mathbf{P}}^{tea}$ are fed into a softmax to be converted to probability distributions before CE to implement probabilistic pseudo mapping learning. Besides, an illustrative example of generating the probabilistic pseudo mapping matrix is provided in Figure 2.

**Optimization.** Finally, we minimize the following combined loss function (final objective) to optimize the student model in an end-to-end training manner:

$$\min_{\theta_{stu}} \ \mathcal{L}_a + \lambda \mathcal{L}_u \quad (15)$$

where $\lambda$ is a ramp-up weighting coefficient used to weight between the supervised alignment learning (i.e., $\mathcal{L}_a$) and pseudo mappings learning (i.e., $\mathcal{L}_u$). In the beginning, the optimization is dominated by $\mathcal{L}_a$ and during the ramp-up period, $\mathcal{L}_u$ will gradually participate in the training to provide more alignment signals. The overall optimization process is outlined in Algorithm 1 (Appendix A), where the student model and the teacher model are updated alternately, and the final student model is utilized for EA inference (Eq. (1)) on test data.

# 5 Experimental Setup

## 5.1 Data and Evaluation Metrics

We evaluate our method on the 15K benchmark dataset (V1) in OpenEA (Sun et al., 2020b) since

the entities thereof follow the degree distribution in real-world KGs. The brief information of experimental data is shown in Table 3 (Appendix B). It contains two cross-lingual settings, i.e., EN-FR-15K (English-to-French) and EN-DE-15K (English-to-German), and two monolingual settings, i.e., D-W-15K (DBPedia-to-Wikidata) and D-Y-15K (DBPedia-to-YAGO). Following the data splits in OpenEA, we use the same split setting where 20%, 10%, and 70% pre-aligned mappings are utilized for training, validation, and testing, respectively.

Entity alignment is a typical ranking problem, where we obtain a target entity ranking list for each source entity by sorting the similarity scores in descending order. We use *Hits@k* (k=1, 5) and *Mean Reciprocal Rank* (*MRR*) as the evaluation metrics (Sun et al., 2020b; Xin et al., 2022a). *Hits@k* is to measure the alignment accuracy, while *MRR* measures the average performance of ranking over all test samples. The higher the *Hits@k* and *MRR*, the better the alignment performance.

## 5.2 Baseline Methods

We choose the methods from the related work as baselines and divide them into two classes below:

- **Structure-based methods.** These methods focus on capturing useful structural context to enrich entity representation, such as (1) *MTransE* (Chen et al., 2017), (2) *GCN-Align* (Wang et al., 2018), (3) *AliNet* (Sun et al., 2020a), (4) *KE-GCN* (Yu et al., 2021) and (5) *IMEA* (Xin et al., 2022a).

- **Semi-supervised methods.** These methods aim to explore alignment signals from unlabeled entities, such as (1) *IPTransE* (Zhu et al., 2017), (2) *SEA* (Pei et al., 2019), (3) *KDCoE* (Chen et al., 2018), (4) *BootEA* (Sun et al., 2018), (5) *MRAEA* (Mao et al., 2020), (6) *RANM* (Cai et al., 2022) and (7) *GAEA* (Xie et al., 2023).

As our method and the above baselines only contain a single model and mainly rely on structural information, for a fair comparison, we do not compare with ensemble-based frameworks (e.g., *CycTEA* (Xin et al., 2022b)) and models infusing side information from multi-modality (e.g., *EVA* (Liu et al., 2021), *RoadEA* (Sun et al., 2022)). For the baseline *RANM*, we remove the name channel to guarantee a fair comparison.

## 5.3 Implementation Details

All the experiments are performed in *PyTorch* on an *NVIDIA GeForce RTX 3090* GPU. Following

| Models | EN-FR-15K | | | EN-DE-15K | | | D-W-15K | | | D-Y-15K | | |
|---|---|---|---|---|---|---|---|---|---|---|---|---|
| | Hit@1 | Hit@5 | MRR | Hit@1 | Hit@5 | MRR | Hit@1 | Hit@5 | MRR | Hit@1 | Hit@5 | MRR |
| MTrasnE[†] | .247 | .467 | .351 | .307 | .518 | .407 | .259 | .461 | .354 | .463 | .675 | .559 |
| GCN-Align[†] | .338 | .589 | .451 | .481 | .679 | .571 | .364 | .580 | .461 | .465 | .626 | .536 |
| AliNet[‡] | .364 | .597 | .467 | .604 | .759 | .673 | .440 | .628 | .522 | .559 | .690 | .617 |
| KE-GCN[‡] | .408 | .670 | .524 | .658 | .822 | .730 | .519 | .727 | .608 | .560 | .750 | .644 |
| IMEA[‡] | .458 | .720 | .574 | .639 | .827 | .724 | .527 | .753 | .626 | .639 | .804 | .712 |
| IPTransE[†] | .169 | .320 | .243 | .350 | .515 | .430 | .232 | .380 | .303 | .313 | .456 | .378 |
| SEA[†] | .280 | .530 | .397 | .530 | .718 | .617 | .360 | .572 | .458 | .500 | .706 | .591 |
| KDCoE[†] | .581 | .680 | .628 | .529 | .629 | .580 | .247 | .412 | .325 | .661 | .764 | .710 |
| BootEA[†] | .507 | .718 | .603 | .675 | .820 | .740 | .572 | .744 | .649 | .739 | .849 | .788 |
| MRAEA[∗] | .537 | .779 | .642 | .681 | .844 | .754 | .574 | .772 | .661 | .725 | .838 | .769 |
| RANM | .567∗ | .780∗ | .663∗ | .686° | .835° | .751° | - | - | - | - | - | - |
| GAEA° | .486 | .746 | .602 | .684 | .854 | .760 | .562 | .768 | .654 | .608 | .791 | .688 |
| MixTEA | **.582** | **.807** | **.680** | **.724** | **.877** | **.797** | **.647** | **.832** | **.731** | **.748** | **.871** | **.802** |
| std. | ±.004 | ±.006 | ±.004 | ±.003 | ±.005 | ±.003 | ±.006 | ±.004 | ±.005 | ±.004 | ±.002 | ±.003 |

Table 1: Entity alignment performance of different methods in the cross-lingual and monolingual settings of OpenEA. The results with † are retrieved from Sun et al. (2020b), and ‡ from Xin et al. (2022a). Results labeled by ∗ are reproduced using the released source codes and labeled by ° are reported in the corresponding references. The **boldface** indicates the best result of each column and underlined the second-best. *std.* means standard deviation.

| Models | EN-FR-15K | | | EN-DE-15K | | |
|---|---|---|---|---|---|---|
| | Hit@1 | Hit@5 | MRR | Hit@1 | Hit@5 | MRR |
| w/o $rel.$ | .471 | .732 | .586 | .679 | .839 | .749 |
| w/o $\mathcal{L}_u$ | .485 | .716 | .589 | .671 | .836 | .744 |
| w/o BDV | .556 | .781 | .656 | .707 | .863 | .777 |
| w/o MDR | .560 | .791 | .663 | .711 | .863 | .779 |
| w/o B&M | .542 | .771 | .644 | .694 | .857 | .766 |
| MixTEA | **.582** | **.807** | **.680** | **.724** | **.877** | **.797** |

| Models | D-W-15K | | | D-Y-15K | | |
|---|---|---|---|---|---|---|
| | Hit@1 | Hit@5 | MRR | Hit@1 | Hit@5 | MRR |
| w/o $rel.$ | .552 | .756 | .641 | .700 | .841 | .761 |
| w/o $\mathcal{L}_u$ | .520 | .708 | .605 | .541 | .684 | .605 |
| w/o BDV | .629 | .812 | .709 | .725 | .856 | .783 |
| w/o MDR | .635 | .822 | .718 | .731 | .862 | .788 |
| w/o B&M | .609 | .802 | .693 | .712 | .849 | .774 |
| MixTEA | **.647** | **.832** | **.731** | **.748** | **.871** | **.802** |

Table 2: Ablation test results.

OpenEA (Sun et al., 2020b), we report the average results of five-fold cross-validation. The embedding dimensions of entities $d_e$ and relations $d_r$ are set to 256 and 128, respectively, the number of GAT layer $L$ is 2, the margin $\rho$ is 2.0, and the momentum $m$ is 0.9. In the EA inference phase, we use Cosine distance as the distance metric and apply Faiss[2] to perform NN search efficiently. The default alignment direction is from left to right, e.g., in D-W-15K, we regard DBpedia as the source KG and seek to find the counterparts of source entities in the target KG Wikidata. The details of hyperparameter settings are shown in Appendix C.

[2] https://github.com/facebookresearch/faiss

# 6 Experimental Results

## 6.1 Performance Comparison

Table 1 reports the experimental results of all the methods on the OpenEA 15K datasets. Even utilizing only the structure information, KE-GCN and IMEA achieve inspiring performance by exploring the rich structural contexts. However, they are hard to further improve the performance because suffer from the lack of enough training data. We also observe that some semi-supervised EA methods (e.g., IPTransE and SEA) fail to outperform these structure-based EA methods, reflecting the fact that both encoder design and semi-supervised strategy are important components of facilitating high-accuracy EA. IPTransE obtains unsatisfactory alignment results since it produces many noise pseudo mappings during the self-training process but does not design an appropriate mechanism to eliminate the influence of noise. Besides, the performance of KDCoE is unstable. According to Sun et al. (2020b), this is because many entities lack textual descriptions, thus preventing the model from finding complementary mappings for co-training from the textual description embedding model. BootEA and MRAEA are competitive baselines in semi-supervised EA domain. Nevertheless, BootEA needs a carefully fine-tuned confidence threshold to filter pseudo mappings, which often leads to unstability, while MRAEA and RANM still follows the data augmentation paradigm, which ignores the uncertainty of pseudo mappings and

is prone to cause error accumulation. Although GAEA learns representations of vast unseen entities via contrastive learning, its performance is unstable. The bottom part of Table 1 shows our method consistently achieves the best performance in all tasks with a small standard deviation (*std.*). More precisely, our model surpasses state-of-the-art baselines averagely by 3.1%, 3.3%, and 3.5% in terms of *Hit@1*, *Hit@5*, and *MRR*, respectively.

## 6.2 Ablation Study

To verify the effectiveness of our method, we perform the ablation study with the following variant settings: (1) w/o $rel.$ removes the relation modeling. (2) w/o $\mathcal{L}_u$ removes probabilistic pseudo mapping learning. (3) w/o BDV only considers EA decisions in the default alignment direction to generate pseudo mappings instead of applying the bi-directional voting strategy (i.e., $\mathbf{P}^{tea} = g(\mathbf{M}^{tea}_{s \to t})$). (4) w/o MDR removes matching diversity-based rectification module in pseudo mappings learning. (5) w/o B&M denotes that the complete model deletes both BDV and MDR module.

The ablation results are shown in Table 2. We can observe that the complete model achieves the best experimental results, which indicates that each component in our model design contributes to the performance improvement. Removing relation modeling from entity representation causes performance drops, which identifies the relation semantics can help in enriching the expressiveness of entity representations. W/o $\mathcal{L}_u$ caused the most significant performance degradation, especially in monolingual settings, showing the crucial role of the pseudo mapping learning in general. The results of w/o BDV and w/o MDR suggest that the bi-directional voting strategy and matching diversity-based rectification module can do benefit to improving the quality of pseudo mapping learning. W/o B&M also demonstrates that the combination of BDV and MDR can further improve the alignment performance. Although w/o BDV only takes EA decisions in one direction into account, it still inevitably brings matching conflicts since NN search neglects the inter-dependency between different EA decisions. Compared to w/o B&M, the MDR in w/o BDV has a certain positive effect, which indicates that our proposed MDR can be applied to other pseudo mapping generation algorithms and help the models to train better.

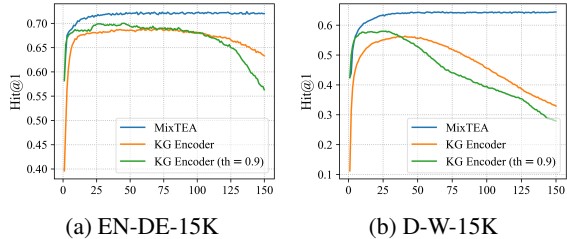

(a) EN-DE-15K     (b) D-W-15K

Figure 3: Test *Hit@1* curve throughout training epochs.

## 6.3 Auxiliary Experiments

**Training visualization.** To inspect our method comprehensively, we also plot the test *Hit@1* curve throughout the training epochs in Figure 3. KG Encoder (th=0.9) represents the KG Encoder described in Sec. 4.1 applying the self-training with threshold=0.9 to generate pseudo mappings every 20 epochs. We control the same experimental settings to remove the performance perturbations induced by different parameters. From Figure 3, we observe that our method converges quickly and achieves the best and most stable alignment performance. The performance of the KG Encoder gradually decreases in the later stages since it gets stuck in overfitting to the limited training data. Although self-training brings some performance gains after data augmentation in the early stages, the performance drops dramatically in the later stages. This is because it involves many noise pseudo mappings and causes error accumulation as the self-training continues. In the later stages, self-training has difficulty in further generating new mappings while existing erroneous mappings constantly misguide the model training, thus hurting the performance.

**Hyperparameter analysis.** We design hyperparameter experiments to investigate the performance varies with some hyperparameters. Due to the space limitation, these experimental results and analyses are listed in Appendix D.

## 7 Conclusion

In this paper, we propose a novel semi-supervised EA framework, termed as MixTEA, which guides the model learning with an end-to-end mixture teaching of manually labeled mappings and probabilistic pseudo mappings. Meanwhile, we propose a bi-directional voting (BDV) strategy and a matching diversity-based rectification (MDR) module to assist the probabilistic pseudo mapping learning. Experimental results on benchmark datasets show the effectiveness of our proposed method.

## Limitations

Although we have demonstrated the effectiveness of MixTEA, there are still some limitations that should be addressed in the future: (1) Currently, we only utilize structural contexts which are abundant and always available in KGs to embed entities. However, when side information (e.g., visual contexts, literal contexts) is available, MixTEA needs to be extended into a more comprehensive EA framework and ensure that it does not become over-complex in the teacher-student architecture. Therefore, how to involve this side information is our future work. (2) Vanilla self-training iteratively generates pseudo mappings and adds them to the training data, where the technicians can perform spot checks during model training to monitor the quality of pseudo mappings. While MixTEA computes probabilistic pseudo mapping matrix and performs end-to-end training, thus making it hard to provide explicit entity mappings for the technicians to check their correctness. Therefore, it is imperative to design a strategy to combine the self-training and probabilistic pseudo mapping learning to enhance the interpretability and operability.

## Ethics Statement

This work does not involve any discrimination, social bias, or private data. Therefore, we believe that our study complies with the ACL Ethics Policy.

## Acknowledgments

We thank the anonymous reviewers for their comments. This work is supported by the National Natural Science Foundation of China No. 62172428.

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

# A  Pseudocode of Training Procedure

**Algorithm 1** Training Procedure
***
**Input:** Knowledge graphs $\mathcal{G}_s$ and $\mathcal{G}_t$; pre-aligned entity mappings $\mathcal{S}$; unlabeled entity set $\hat{\mathcal{E}}_s$ and $\hat{\mathcal{E}}_t$; momentum $m$; margin $\rho$.
**Output:** the student encoder parameters $\theta_{stu}$.
1: Initialize entity embeddings and relation embeddings;
2: Initialize teacher model $\theta_{tea}$ and student model $\theta_{stu}$;
3: **for** each epoch **do**
4:    Encode entities using $f(;\theta_{stu})$ and $f(;\theta_{tea})$;
5:    Calculate $\mathcal{L}_a$ by supervised learning in Eq. (8);
6:    Generate pseudo mapping matrix via BDV strategy;
7:    Rectify pseudo mapping matrix via MDR module;
8:    Calculate $\mathcal{L}_u$ by pseudo mapping learning in Eq. (14);

9:    $\theta_{stu} \leftarrow \text{BackProp}(\mathcal{L}_a + \lambda\mathcal{L}_c)$;      ▷Adam Update
10:   $\theta_{tea} \leftarrow m\theta_{tea} + (1-m)\theta_{stu}$;
11: **end for**
12: **return** the student encoder parameters $\theta_{stu}$
***

# B  Dataset Statistics

| Datasets | | #Ent. | #Rel. | #Tri. |
|---|---|---|---|---|
| EN-FR-15K | English | 15000 | 267 | 47334 |
| | French | 15000 | 210 | 40864 |
| EN-DE-15K | English | 15000 | 215 | 47676 |
| | German | 15000 | 131 | 50419 |
| D-W-15K | DBPedia | 15000 | 248 | 38265 |
| | Wikidata | 15000 | 169 | 42746 |
| D-Y-15K | DBPedia | 15000 | 165 | 30292 |
| | YAGO | 15000 | 28 | 26638 |

Table 3: Dataset statistics: #Ent., #Rel., and #Tri. means the number of entities, relations, and triples in corresponding KG, respectively.

# C  Hyperparameter Details

We tune the hyperparameters for our proposed Mix-TEA. The setting values and search ranges of hyperparameters are described in Table 4.

| Hyperparameters | Value/Search range |
|---|---|
| The number of GAT layer | [1, 2, 3, 4] |
| Momentum parameter $m$ | [0.9, 0.99, 0.999] |
| Margin $\rho$ | [1, 2, 3] |
| Negative sample size | [10, 20, 30] |
| Embedding dimension | [128, 256] |
| Embedding initialization | Xavier |
| Learning rate | 0.005 |

Table 4: Hyperparameter values and search ranges.

# D  Hyperparameter Analysis

**The impact of different GAT layers.** We vary the number of GAT layer $L$ from 1 to 4 and the

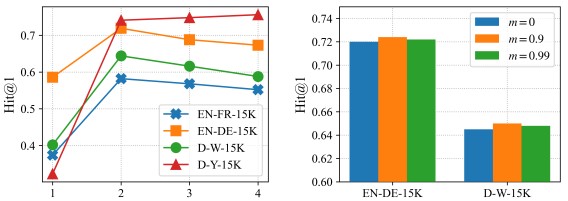

(a) the number of GAT layers    (b) momentum parameter

Figure 4: Sensitivity analysis of our proposed method.

quantitative results are illustrated in Figure 4 (a). $L$=1 results in poor alignment performance due to the limited structural modeling power. The best performance is achieved when $L$=2, except for the D-Y-15K task. Increasing $L$ will not bring further performance improvement, we infer that there is overfitting or oversmoothing during neighborhood aggregation. In D-Y-15K, the optimal performance is obtained when $L$=4. The possible reason is that in D-Y-15K, the two KGs have relatively sparse structure information (as shown in Table 3 in Appendix B), therefore they need to capture more alignment evidence from distant neighbors.

**The impact of momentum parameter.** We investigate the momentum parameter $m$ in [0, 0.9, 0.99] and the results in EN-DE-15K and D-W-15K tasks are presented in Figure 4 (b). We found that our method maintains good performance under different momentum settings, demonstrating that our method is insensitive to $m$. In addition, a proper $m$ such as 0.9 brings certain performance improvements, which indicates that the EMA update manner can facilitate the teacher model to yielding stable and robust pseudo mappings.