# OpenReview forum: "MixTEA: Semi-supervised Entity Alignment with Mixture Teaching"
_EMNLP/2023/Conference — EMNLP 2023 Findings_

### Official Review · Reviewer_KV6A · 2023-08-04

**Soundness:** 3

**Excitement:**

3: Ambivalent: It has merits (e.g., it reports state-of-the-art results, the idea is nice), but there are key weaknesses (e.g., it describes incremental work), and it can significantly benefit from another round of revision. However, I won't object to accepting it if my co-reviewers champion it.

**Missing References:**

No

**Paper Topic And Main Contributions:**

This paper introduces a new method for semi-supervised entity alignment based on Teacher-Student architecture, termed MixTEA. The main contribution of this paper is that the proposed method trains alignment learning with both manually labeled mappings and probabilistic pseudo mappings to alleviate the negative influence of noisy or uncertain pseudo mappings.

**Questions For The Authors:**

See my review.

**Reasons To Accept:**

This paper studies the entity alignment problem in a semi-supervised scenario which is a practical and challenging task. Existing methods mainly suffer from two problems: uncertainty of pseudo mappings and noisy pseudo-mapping learning. To alleviate the negative influence of these two problems, the author proposed a novel method that guides model learning with an end-to-end mixture teaching of manually labeled mappings and probabilistic pseudo mappings. In this part of the experiment, the author conducts extensive experiments to demonstrate the effectiveness of this method.

-  author conducts extensive experiments to demonstrate the effectiveness of this method.
-  The structure of the paper is reasonable and the content is sufficient.
-  The schematic diagram is concise. Readers can fully understand the method proposed in the article through the schematic diagram

**Reasons To Reject:**

- Paper proposes a bi-directional voting (BDV) strategy and a matching diversity-based rectification (MDR) module to assist the probabilistic pseudo-mapping learning but lacks the theoretical demonstration of the effectiveness of the BDV strategy and MDR model.

- I would like to see how the BDV strategy improves conventional alignment strategies.

- In addition, the effectiveness of the MDR module shown in Table 2 is not obvious, this component may be not critical.

- Experiments do not concretely show the effectiveness of your method on alleviating the negative impact of uncertainty of pseudo mappings and noisy pseudo mapping learning.

**Reproducibility:**

4: Could mostly reproduce the results, but there may be some variation because of sample variance or minor variations in their interpretation of the protocol or method.

**Reviewer Confidence:**

4: Quite sure. I tried to check the important points carefully. It's unlikely, though conceivable, that I missed something that should affect my ratings.

---

> ### Author Rebuttal · Authors · 2023-08-28
>
> We thank you for your generous and critical comments on our submitted manuscript. We give the respective explanations and responses as follows:
> 1. In this paper, we introduce probabilistic pseudo mapping modeling into semi-supervised entity alignment, which trains the model with an end-to-end mixture teaching of manually labeled mappings and probabilistic pseudo mappings, unlike previous methods of appending pseudo mappings to the training set iteratively. Our proposed method has the theoretical basis of previous research. To be specific, we select Mean Teacher as the semi-supervised learning framework, which is a stable architecture based on a moving average strategy. Regarding BDV, which is based on the asymmetric nature of alignment directions in EA. Although relevant papers have studied bi-directional strategy before, they only use it for obtaining the final alignment results and did not consider weighted ensemble results from different directions to model the uncertainty (probability) of pseudo mapping. Meanwhile, MDR mitigates the problem of false pseudo mappings by punishing hub entities that align with many other entities and reducing the probability of aligning with these hub entities.
> 2. How to apply BDV to traditional entity alignment methods is a very interesting issue. Firstly, we understand that the traditional alignment strategies you mentioned may refer to the traditional alignment method based on equivalence reasoning or symbol feature similarity. However, our method is neural embedding-based, which dominates current EA research and we have already introduced it in the first section. Traditional alignment methods are usually based on statistics or symbol similarity to obtain the final result, without involving any iterative training or learning process actually. The role of BDV is to generate probability pseudo mappings in each iteration to gradually improve the model performance. Therefore, it cannot be applied to traditional methods directly.
> 3. We think the MDR module is simple and critical. Although the gains from MDR are modest, the results on all four datasets show that it could steadily make contributions. Additionally, MDR is proposed to mitigate false pseudo mappings that are easy to occur during the entity alignment process yet difficult to detect.
> 4. In addition to the comparison experiments to prove the overall effectiveness, and the ablation experiments to prove the effectiveness of each module, we attempted to demonstrate that our model improves the alignment performance and pseudo mapping quality during the iteration process in section 6.3-"Training visualization". In 6.3, we compared our model with a variant that uses a threshold to filter pseudo mappings. The test results on every epoch are shown in Figure 3. As we can see in Figure 3, the method applying threshold filtering (green line) will result in performance degradation in the later stages of training, which is due to the selection of too many wrong pseudo mappings in the early stage, resulting in the error accumulation phenomenon. In contrast, our model (blue line) which applies BDV and MDR to optimize the process of pseudo mapping learning was able to produce optimal and stable performance. This indicates that incorrect pseudo mappings in the early stage would gradually corrected by BDV and MDR during the end-to-end training process, thereby avoiding negative effects.

---

### Official Review · Reviewer_ce7N · 2023-08-04

**Soundness:** 2

**Excitement:**

2: Mediocre: This paper makes marginal contributions (vs non-contemporaneous work), so I would rather not see it in the conference.

**Paper Topic And Main Contributions:**

This paper proposes a semi-supervised EA method, which guides the model learning with an end-to-end mix-ture teaching of manually labeled mappings and probabilistic pseudo mappings.


**Questions For The Authors:**

1. Why BSD works?
2. Why MDR works? What is the meaning of Equation (12)?

**Reasons To Accept:**

1. This authors introduce a bi-directional voting (BDV) strategy which utilizes the alignment decisions in different directions to estimate the uncertainty of pseudo mappings.
2. This authors design a matching diversity-based rectification (MDR) module to adjust the pseudo mapping learning.
3. Experimental results seem good.


**Reasons To Reject:**

1. This paper is not well motivated
2. The proposed BDV and MDR are not clearly explained


**Reproducibility:**

4: Could mostly reproduce the results, but there may be some variation because of sample variance or minor variations in their interpretation of the protocol or method.

**Reviewer Confidence:**

3: Pretty sure, but there's a chance I missed something. Although I have a good feel for this area in general, I did not carefully check the paper's details, e.g., the math, experimental design, or novelty.

---

> ### Author Rebuttal · Authors · 2023-08-28
>
> Thank you for your comment. However, we fully disagree with you respectfully.
> First of all, the motivation of our paper is clear, which is also appreciated by other reviewers. In this paper, we focus on tackling the two problems in pseudo mapping learning of semi-supervised entity alignment, i.e., (1) noisy pseudo mapping learning and (2) modeling uncertainty of pseudo mappings, and we propose our MixTEA to handle the above problems.
> 1. Why BDV works? BDV module is the key to modeling the uncertainty of pseudo mapping in this paper. Typically, previous entity alignment methods use the nearest neighbor (NN) search to obtain its counterpart for each entity. However, NN search in different directions will yield asymmetric results because the entities in an entity pair may not be each other's nearest neighbors (i.e. e_1's nearest neighbor is e_a, but e_a's nearest neighbor may be e_2). Therefore, the asymmetric nature of alignment directions in EA inspires us to weighted ensemble alignment results from different directions. In this way, we not only obtain more comprehensive alignment results (the results come from two directions), but also model the uncertainty of pseudo mappings.
> 2. Why MDR works? MDR module can mitigate the problem of false (erroneous) pseudo mappings by reducing the confidence of alignment associated with hub entities. A Hub entity is an entity that aligns with many other entities, i.e., a hub entity repeatedly occurs as the nearest neighbors of others, which easily happens in the high-dimension vector space (also called hubness problem). However, the ideal prediction of entity alignment is one-to-one matching, which means a source entity can be matched with at most one target entity, and vice versa. As a result, hub entities will cause many false pseudo mappings in semi-supervised learning. MDR is a rectification strategy to punish the hub entities. In Eq (12), the numerator is the confidence of the alignment pair, and the denominator is the sum of the confidence that the entities in the entity pair align with other entities. Given an alignment pair (e_1, e_a), if e_a is the hub entity, the denominator in Eq (12) will bigger than numerator, and the original confidence of the alignment pair (e_1, e_a) will be reduced (i.e., penalized) because e_a is a hub entity. But if (e_1, e_a) is the true pseudo mapping, then the denominator and numerator are equal in Eq (12).

---

### Official Review · Reviewer_8XC3 · 2023-08-05

**Soundness:** 3

**Excitement:**

3: Ambivalent: It has merits (e.g., it reports state-of-the-art results, the idea is nice), but there are key weaknesses (e.g., it describes incremental work), and it can significantly benefit from another round of revision. However, I won't object to accepting it if my co-reviewers champion it.

**Paper Topic And Main Contributions:**

The authors propose a new end-to-end method MixTEA for a semi-supervised entity alignment task. MixTEA has a teacher-student architecture, where both student and teacher models are knowledge graph encoders. The method is claimed to overcome two significant issues of semi-supervised EA methods: uncertainty of pseudo mapping (by switching a binary pseudo mapping to a probabilistic one and estimating the matching uncertainty via confidence score) and an increased impact of noisy pseudo mappings (by a matching diversity-based rectification).



**Reasons To Accept:**

- The paper addresses the challenging problem of using unlabeled data for entity alignment. It is well-motivated and clearly written.
- The paper proposes several novel and interesting approaches, such as BDV (bi-directional voting for more comprehensive pseudo mappings) and MDR (matching diversity-based rectification for dynamic mitigating the influence of noisy mappings).
- The authors provide an extensive and detailed ablation study and results discussion.


**Reasons To Reject:**

- Some inaccuracies in mathematical notation (e.g., line 278 - what is "l" in "l-1"?).
- In Section 4, the authors claim that the end-to-end training "...gradually improves the quality of pseudo mapping". It would be useful to add some example/analysis/proof that this is indeed happening and the mappings are becoming more and more accurate (e.g., the results of the model after every n-th iteration?).
- Lines 329-333: being an important part of the algorithm, this MixTEA part is not reflected in Figure 1.
- The experimental setup remains unclear to me. Was any additional unlabeled data used? How were the hyperparameters tuned? What was the search range?
- No standard deviation for the baselines. Note that most of the baseline values are taken directly from the previous work, and some models have been trained in other settings. Therefore, a comparison of MixTEA with the baseline models should be done more carefully.
- Figure 3 is not quite clear.
- Section 6.1: the threshold values are mentioned here for the first time and are not discussed elsewhere. I guess the whole setting could be strengthened by adding more sophisticated methods for threshold tuning (e.g., a recent ACL paper [1]); apart from that, at least some discussion of the threshold values is required.

[1] Sedova and Roth. 2023. "ACTC: Active Threshold Calibration for Cold-Start Knowledge Graph Completion"

**Reproducibility:**

3: Could reproduce the results with some difficulty. The settings of parameters are underspecified or subjectively determined; the training/evaluation data are not widely available.

**Reviewer Confidence:**

5: Positive that my evaluation is correct. I read the paper very carefully and I am very familiar with related work.

**Typos Grammar Style And Presentation Improvements:**

- line 61, 64, 418, 545.. - unnecessary brackets in citations
- line 533: "enough" should be removed -> "the lack of training data"

---

> ### Author Rebuttal · Authors · 2023-08-28
>
> We thank you for your generous and critical comments on our submitted manuscript. We give the respective explanations and responses as follows:
> 1. "l" means "l-th" layer in the multi-layer graph attention network (GAT), which has been explained in Line 287. GAT can be stacked in multiple layers, and each layer can capture neighbor structure information with different hops. Eq (12) is the layer-wise propagation rule.
> 2. We attempted to demonstrate that our model improves the alignment performance and pseudo mapping quality during the iteration process in section 6.3-"Training visualization".  In 6.3, we compared our model with a variant that uses a threshold to filter pseudo mappings. The test results on every epoch are shown in Figure 3. As we can see in Figure 3, the method applying threshold filtering (green line) will result in performance degradation in the later stages of training, which is due to the selection of too many wrong pseudo mappings in the early stage, resulting in the error accumulation phenomenon. In contrast, our model (blue line) which applies BDV and MDR to optimize the process of pseudo mapping learning was able to produce optimal and stable performance. This indicates that incorrect pseudo mappings in the early stage would gradually corrected by BDV and MDR during the end-to-end training process, thereby avoiding negative effects.
> 3. Lines 329 to 333 mainly describe the pseudo mapping learning provided by the teacher model which has been drawn in the bottom right corner of Figure 1. In the subsequent sub-sections, we also provided a detailed introduction based on the bottom right corner of Figure 1.
> 4. We are sorry for the confusion caused by the incomplete description of the experimental details. Our experimental setup strictly follows the semi-supervised setting in OpenEA, and we use unlabeled data provided in OpenEA as additional data for semi-supervised learning. At the same time, we have discussed some hyper-parameters (GAT number and momentum parameter) of the model in Appendix (C), and we will supplement more parameter experiments later. As for the parameter tuning, the layer number of GAT is searched in [1,2,3,4], the momentum parameter is set to [0.9,0.99,0.999], the margin is searched from [1,2,3], and the hidden dimension is selected in [128, 256]. We attach great importance to the issue of insufficient experimental details and will carefully supplement them in the manuscript.
> 5. For a fair comparison, most of the baseline values are taken directly from their corresponding papers and our method is also under the same experimental settings with baselines. In this paper, we also give the standard deviation in Table 1 to demonstrate the stability of our model under five-fold cross-validation.
> 6. To inspect our method comprehensively, we also plot the test Hit@1 curve throughout the training phase in Figure 3. To be specific, after every training epoch, we test the model over the test dataset and record the Hit@1 scores. So the x-axis of Figure 3 means epoch number and the y-axis means Hit@1 score. We are sorry that the figure does not clearly indicate the x-axis information, which has made you feel confused. Figure 3 shows that our model (blue line) achieves the best and most stable performance.
> 7. Our proposed method does not require any threshold selection. The threshold we mentioned in 6.3 is the setting of the variant model, i.e., KG Encoder (th=0.9). The threshold selection for the compared variant model is indeed empirical, and we choose the threshold equal to 0.9, which is a relatively high confidence level to filter out low-confidence mappings. Perhaps in future work, we can take inspiration from this threshold-tuning strategy, but our currently proposed method does not require a threshold.

---

### Meta-Review · Area_Chair_X2VV · 2023-09-19

**Recommendation:** 3

**Metareview:**

The authors propose a method, MixTEA, for semi-supervised entity alignment based on a Teacher-Student architecture that works by training the student model using alignment signals from both manually labeled mappings and probabilistic pseudo mappings generated by the teacher model. Compared to existing works, the claimed methodological innovation mitigates issues associated with noisy pseudo mappings and/or lack of pseudo mapping uncertainty consideration in utilizing the pseudo labels.

== Quality ==
The reviewers believed the methodological innovations were clearly described and the paper was clearly written overall. This is a widely studied problem and the authors propose a conceptual improvement to account for uncertainly and operationalize it in a sensible way with a teacher-student model. Finally, the authors provide an extensive comparison relative to competing EA methods and a good ablation study -- where the results are good overall and establish a state-of-the-art solution under realistic constraints.

== Clarity ==
As stated in the 'quality' section, the paper is well-written overall. Additionally, the authors make good use of figures, etc. to better clarify methodological explanations. However, there were also some concerns regarding attributing empirical improvements to the specific methods proposed. Specifically, even after rebuttal and discussion, it still wasn't clear to the reviewers after discussion how much the empirical improvements were due to the proposed methods and how much was due to general empirical improvements in working to outperform a known target (i.e., hyperparameter optimization). This may be able to be remedied with more details of the experimental setup and further discussion of results. Clearly, additional theoretical understanding of BDV and MDR based improvements would be useful (or references to support this from other settings), but minimally a more detailed empirical analysis is necessary to establish their relative contributions to the the improvement of the models. This was partially addressed in the rebuttal, but the reviewers remained unconvinced that the authors have made a strong case regarding the specific empirical contributions of the proposed improvements.

== Originality ==
The paper proposes multiple novel, interesting, and sensible approaches, such as BDV (bi-directional voting for more comprehensive pseudo mappings) and MDR (matching diversity-based rectification for dynamic mitigating the influence of noisy mappings). For the most part, these innovations are specific to EA and build on existing baselines and datasets. However, there is more conceptual innovation than many EA papers.

== Significance ==
This is a widely studied problem where performance improvements have practical and potentially impactful implications. Within the EA-specific community, this work will very likely be used as a baseline for future comparisons. That being said, this work builds on existing experimental setups and datasets, so it is primarily a methods contribution. Additionally, the proposed methods to provide support for teacher-student models as an uncertainty modeling technique, but in general isn't clearly going to influence work in other tasks. Thus, it is a solid work within this application.

---

### Decision · Program_Chairs · 2023-10-07

**Decision:**

Accept-Findings

**Comment:**

The authors propose a method, MixTEA, for semi-supervised entity alignment based on a Teacher-Student architecture that works by training the student model using alignment signals from both manually labeled mappings and probabilistic pseudo mappings generated by the teacher model. Compared to existing works, the claimed methodological innovation mitigates issues associated with noisy pseudo mappings and/or lack of pseudo mapping uncertainty consideration in utilizing the pseudo labels.

== Quality ==
The reviewers believed the methodological innovations were clearly described and the paper was clearly written overall. This is a widely studied problem and the authors propose a conceptual improvement to account for uncertainly and operationalize it in a sensible way with a teacher-student model. Finally, the authors provide an extensive comparison relative to competing EA methods and a good ablation study -- where the results are good overall and establish a state-of-the-art solution under realistic constraints.

== Clarity ==
As stated in the 'quality' section, the paper is well-written overall. Additionally, the authors make good use of figures, etc. to better clarify methodological explanations. However, there were also some concerns regarding attributing empirical improvements to the specific methods proposed. Specifically, even after rebuttal and discussion, it still wasn't clear to the reviewers after discussion how much the empirical improvements were due to the proposed methods and how much was due to general empirical improvements in working to outperform a known target (i.e., hyperparameter optimization). This may be able to be remedied with more details of the experimental setup and further discussion of results. Clearly, additional theoretical understanding of BDV and MDR based improvements would be useful (or references to support this from other settings), but minimally a more detailed empirical analysis is necessary to establish their relative contributions to the the improvement of the models. This was partially addressed in the rebuttal, but the reviewers remained unconvinced that the authors have made a strong case regarding the specific empirical contributions of the proposed improvements.

== Originality ==
The paper proposes multiple novel, interesting, and sensible approaches, such as BDV (bi-directional voting for more comprehensive pseudo mappings) and MDR (matching diversity-based rectification for dynamic mitigating the influence of noisy mappings). For the most part, these innovations are specific to EA and build on existing baselines and datasets. However, there is more conceptual innovation than many EA papers.

== Significance ==
This is a widely studied problem where performance improvements have practical and potentially impactful implications. Within the EA-specific community, this work will very likely be used as a baseline for future comparisons. That being said, this work builds on existing experimental setups and datasets, so it is primarily a methods contribution. Additionally, the proposed methods to provide support for teacher-student models as an uncertainty modeling technique, but in general isn't clearly going to influence work in other tasks. Thus, it is a solid work within this application.